# Development of Hydroxyapatite Coatings for Orthopaedic Implants from Colloidal Solutions: Part 1—Effect of Solution Concentration and Deposition Kinetics

**DOI:** 10.3390/nano13182577

**Published:** 2023-09-17

**Authors:** Bríd Murphy, Mick A. Morris, Jhonattan Baez

**Affiliations:** 1Advanced Materials & Bioengineering Research Centre (AMBER), Trinity College Dublin, Dublin 2, D02 CP49 Dublin, Ireland; baezj@tcd.ie; 2School of Chemistry, Trinity College Dublin, Dublin 2, D02 PN40 Dublin, Ireland

**Keywords:** hydroxyapatite, solution deposition, novel coating, deposition kinetics, functional coatings, structural characterizations

## Abstract

This study introduces and explores the use of supersaturated solutions of calcium and phosphate ions to generate well-defined hydroxyapatite coatings for orthopaedic implants. The deposition of hydroxyapatite is conducted via several solutions of metastable precursors that precipitate insoluble hydroxyapatite minerals at a substrate–solution interface. Solutions of this nature are intrinsically unstable, but this paper outlines process windows in terms of time, temperature, concentration and pH in which coating deposition is controlled via the stop/go reaction. To understand the kinetics of the deposition process, comparisons based on ionic strength, particle size, electron imaging, elemental analyses and mass of the formed coating for various deposition solutions are carried out. This comprehensive dataset enables the measurement of deposition kinetics and identification of an optimum solution and its reaction mechanism. This study has established stable and reproducible process windows, which are precisely controlled, leading to the successful formation of desired hydroxyapatite films. The data demonstrate that this process is a promising and highly repeatable method for forming hydroxyapatites with desirable thickness, morphology and chemical composition at low temperatures and low capital cost compared to the existing techniques.

## 1. Introduction

Orthopaedic implant fixation is hindered by fibrosis encapsulation, which is the body’s natural response to a foreign object [1,2,3,4]. Coating metallic implants with a bio-compatible layer can mitigate fibrous build-up and promote implant fixation [5,6]. 

Being the main ceramic component of bones and teeth in the human body, with properties of rigidity and compressive strength and a brittle nature, hydroxyapatite (HA) coatings are used to improve the osteointegration of orthopaedic implants. The use of synthetic HA coating aids orthopaedic recovery since it mimics natural bone, and when deposited in a porous structure, it encourages new bone-ingrowth directly onto an implant [7]. HA coating is particularity important for total hip and total knee replacements as it accelerates implant fixation at these load-bearing joints [8].

HA is a crystalline phase of calcium, phosphate and hydroxyl groups of chemical formula Ca_10_(PO_4_)_6_(OH)_2_ and hexagonal crystal structure of space group P6_3_/m. Biological HA is a product of the nucleation and crystallisation of calcium and phosphate ions in vivo, but these ions in solution can also synthetically initiate HA formation. Synthetic HA can enhance bone growth in vivo, and its deposition can be achieved on a wide range of materials [9,10,11,12,13]. Certain types of synthetic HA as doped-HA or HA nanoparticles are being explored for their antimicrobial performance to aid in post-orthopaedic operative infections [14,15]. 

However, synthetic methods of coating metallic parts with HA are far from facile. Solution-based HA processes highlight a requirement to regulate the ionic strength of solutions [16,17,18,19]. Chemical precipitation is advantageous in forming HA nanoparticles and nano-powders because the phases of HA can be controlled during this process, but it has not yet been extensively explored as a coating technique [20,21,22,23,24,25]. Similarly, precipitation of Ca^2+^ and PO_4_^3−^ ions out of a solution onto a surface to form a HA layer has been achieved with relevant precursors under specific conditions of pH and temperature on collagen or bone interfaces [26,27]. In general, formation of a complex crystalline structure with the desired morphology, adhesion properties and mechanical strength remains a significant challenge. Despite a plethora of methods that lead to HA formation, not many replicate the body’s natural process of HA growth, and those that do are limited by poor porosity, phase and thickness control, along with limited substrate adhesion to a relevant orthopaedic implant [28,29].

HA crystallisation in vivo is thought to start with 10–25 nm sized amorphous apatite particles [30]. Studies suggest these initial particles are Posner clusters of Ca_9_(PO_4_)_6_, which are formed from the nucleation of phosphate-based fluid in contact with calcium semi-solid interfaces [31,32,33]. These particles accumulate into clusters and act as a precursor to amorphous calcium phosphate (ACP), with their shape allowing for pores and interstices in the HA as it forms [34,35,36]. Formation of synthetic HA under aqueous conditions should allow for ACP nucleation followed by HA crystal growth which replicates the body’s endogenous bone growth process [37]. However, the formation of HA at surfaces is not well understood; the focus of research has mainly been on characterisation rather than the mechanism of film formation. This study builds upon the findings of the scientific community regarding other HA deposition methods, both biologically and synthetically. With a focus on the required outcomes of porosity, herein, chemical structure analysis is undertaken at various stages to tie the characterisation outcomes back to the process itself.

This method of depositing hydroxyapatite (HA) on orthopaedic implant-type substrates mimics biological-like self-assembly growth processes. One novelty of this process lies in the use of chemical precipitation directly onto a substate to be coated without intermediate steps. Using these saturated solutions of calcium and phosphorous gives rise to coatings with adequate coverage, thickness and morphology control that are relevant to manufacturing requirements. This method also allows for a non-line of sight HA deposition, which is also another of its advantages over the existing techniques [38,39,40]. It is possible to build on the existing literature and studies to hypothesise how a colloidal method might work. At a near biological pH of 7 in aqueous supersaturated conditions, studies show that HA will form via apatitic precursors [41,42]. The precursors formed should follow Ostwald’s rule of 4, whereby the most thermodynamically unstable phase forms first [43]. 

Our results detail a proposed nucleation and growth pathway for HA within this deposition system. With the aim of identifying the ideal process solutions, analysis is carried out on pH, ionic strength, and dynamic light scattering (DLS) properties. Subsequently, HA films are formed with different solutions and are analysed. Weight analysis, scanning electron microscopy (SEM) and energy dispersive X-ray spectroscopy (EDX) of the deposited films reveals the solutions which generated HA films with coverage, morphology and chemical structure best aligned to osetoinduction HA coatings. 

## 2. Materials and Methods

All materials and reagents were used as received. Monobasic potassium phosphate (KH_2_PO_4_) United States Pharmacopeia (USP) reference standard, Honeywell Fluka hydrochloric (HCl) acid solution 6 M, tris(hydroxymethyl)-aminomethane (TRIS) ACS reagent, 99.8% sodium chloride (NaCl) BioXtra, and 99.5% calcium nitrate tetrahydrate (Ca(NO_3_)_2_·4H_2_O) ACS reagent were all purchased from Sigma Aldrich (St. Louis, MO, USA). A calibrated benchtop pH meter with a temperature-enabled probe (Orion Star A111, Thermo Scientific™, Oxford, UK) was used for pH measurements with an accuracy of 0.001 pH. Titanium coupons of Ti-6Al-4V alloy were used as substrates. All substrates were submerged in hot basic solutions to increase roughness and yield a negatively charged surface for calcium ion attachment [44,45,46,47]. 

KH_2_PO_4_, TRIS and NaCl were mixed in deionized water (DIW) to yield a supersaturated phosphate concentrate. See Appendix A for molarities, CAS numbers and water solubility of reagents. HCl was added to increase the stability of the concentrate to prevent precipitation. Ca(NO_3_)_2_·4H_2_O was mixed with DIW to yield a supersaturated calcium concentrate. For deposition process runs, the supersaturated concentrates were combined and diluted to various concentrations.

Solutions of varying dilution are identified with %*v*/*v*, which is the percentage of concentrate volume to process solution volume. Solutions were warmed to 40–50 °C for the deposition of hydroxyapatite. 

Ionic strength is a well-established modifier in hydroxyapatite formation since it increases or decreases the ions present in a solution [48,49]. Ionic strength was calculated as follows. 

Equation (1): Ionic Strength (I) formula: I = ½S((C_KH_2_PO_4__·(Z_KH_2_PO_4__)^2^) + (C_TRIS_·(Z_TRIS_)^2^) + (C_NaCl_·(Z_NaCl_)^2^) + (C_Ca(NO_3_)_2__·(Z_Ca(NO_3_)_2__)^2^)) (1)
where C = mol·L^−1^ and Z = electric charge.

Despite the ratio of components remaining fixed, ionic strength of the supersaturated solutions is increased via sodium chloride addition (increasing the ratio of concentrate in solution); see Table 1 for list of the solution data.

A custom-designed experimental apparatus was used for deposition. A sample holder, a thermometer, and a pH probe for in situ temperature and acidity measurements, and an overhead stirrer to agitate the solution were inserted into the vessel. High agitation rates of 1000 RPM prevented gross precipitation of mineral from the solution. Substrates were placed in the reaction vessel for deposition, and then removed and rinsed with DIW; this process was repeated several times with fresh solutions to grow a coherent layer of HA at the solution–substrate interface of desired thickness. Between deposition steps, substrates were removed from solution and allowed to dry for a minimum of 15 min in ambient conditions (to ensure that active sites are available for subsequent process runs). More data behind the drying rational are provided in Appendix A. To assess the effect of different solutions on eventual film growth, identical substrates were subjected to several deposition process runs of each solution and analysed intermittently, i.e., one substrate underwent 7 deposition runs using solution 1, one substrate underwent 7 deposition runs using solution 2, etc. 

Dynamic Light Scattering (DLS) was performed using Malvern Zetasizer Nano ZS (Worcestershire, UK), with 633 nm HeNe laser. Samples were added to glass cuvette which had a 10 mm pathlength and allowed to equilibrate for 120 s at 47 °C prior to sampling. The machine was operated in backscatter mode at an angle of 173°. Scanning Electron Microscopy (SEM) images were recorded on a Zeiss Ultra Plus system (Jena, Germany) with accelerating voltages in the range of 5–10 kV, at a working distance between 3 and 10 mm and with an in-lens detector. Energy-dispersive X-ray spectroscopy (EDX) spectra were acquired at 15 kV on an Oxford Inca EDX detector (Oxford Instruments, High Wycombe, UK). 

## 3. Results

### 3.1. Kinetics of Process Solutions

The solution reactions are complex but may be thought of using the following reaction schemes which detail the formation of hydroxyapatite (HA). At equilibrium, calcium and phosphate ions in solution form a tricalcium phosphate (TCP) mineral product as shown in Equation (2). 

Equation (2): Equilibrium chemical reaction for the formation of tricalcium phosphate (TCP) from calcium and phosphate ions.
3Ca^2+^_(aq)_ + 2PO_4_^3−^_(aq)_ ⇌ Ca_3_(PO_4_)_2(s)_
(2)

At stoichiometric proportions and in the presence of hydroxyl ions, the formation of TCP is thermodynamically favoured. To form pure hydroxyapatite (HA) under solution, excess calcium ions as Ca^2+^ or Ca(OH)_2_ are required, Equation (3). 

Equation (3): Example of one reaction where additional calcium ions with tricalcium phosphate and hydroxyl ions can form hydroxyapatite (HA).
3Ca_3_(PO_4_)_2(s)_ + Ca^2+^_(aq)_ + 2OH^−^_(aq)_ ⇌ Ca_10_(PO_4_)_6_(OH)_2(s)_
(3)

In this work, as the concentration of solutions are equally supersaturated with respect to calcium and phosphate ions, they drive the precipitation of thermodynamically unstable phases of octacalcium phosphate (OCP), as shown in Equations (4) and (5) [50,51,52].

Equation (4): Formation of calcium-deficient (CDHA) precursor to octacalcium phosphate (OCP) particles in aqueous solution.
9Ca^2+^_(aq)_ + 6PO_4_^3−^_(aq)_ + OH^−^_(aq)_ + H^+^_(aq)_ ⇌ Ca_9_(HPO_4_)(PO_4_)_5_(OH)_(s)_
(4)

Equation (5): Nucleation of octacalcium phosphate (OCP) particle in aqueous solution.
Ca_9_(HPO_4_)(PO_4_)_5_(OH)_(s)_ + 5H_2_O_(aq)_ ⇌ Ca_8_H_2_(PO_4_)_6_·5H_2_O_(s)_ + CaO_(s)_
(5)

Whilst the experimental solutions at room temperature contain metastable intermediaries (which form and then re-dissolve continually), their nucleation into stable precipitates only occurred above a critical temperature, and deposition only occurred between 44 and 50 °C. There is an upper temperature limit because of redissolution. Figure 1a, which collates temperature from more than 100 process runs (including process runs from various solutions), shows a mean of 46.9 °C.

Precipitation of Ca-P phases from solution should be associated with a pH drop as a result of decreased calcium ion concentration [53]. As Equations (1)–(3) show, the formation of HA species initiates a drop in calcium ion concentration and OH^−^ concentration. Within this study, pH dropped by 0.12 ± 0.02 from 0 to 30 min with almost no change after that; therefore, process runs were ceased at 30 min for all solutions, Figure 1b. The pH reduction confirmed that ions are removed from solution, and their concentration decreases with the solution becoming more thermodynamically stable until the rate of deposition becomes kinetically too slow.

While all constituent parts were accounted for in the ionic strength calculations, 95% of the ionic strength value for each solution was due to the NaCl component, see Table 1 for values. 

DLS data from various process solutions shows that the particle size in solution varies significantly with dilution becoming smaller with increased dilution, Figure 1c. Solution of 12%*v*/*v* has particle size with a mean of 5.2 µm, whereas concentrations between 6 and 9%*v*/*v* have a particle size of around 100 nm, and the concentration of 3%*v*/*v* have 35% of particles that are <8 nm. 

### 3.2. Analysis of HA Films Formed with Various Process Solutions

When a substrate is added to the solutions at the correct temperature, heterogeneous nucleation occurs at the substrate, thereby forming a deposit. A side-by-side photograph of samples post 7 deposition runs reveals a colour gradient of the HA film across solutions; see Figure 2. It is clear from this photograph that solutions 5, 6 and 7 have more white material, and we postulate that some of this is just apatite mineral deposited on the surface as it is easily removed. The darker colour gradient seen in solution 3 and 4 indicates that more well adhered crystal growth bonded onto the substrates, as it cannot be removed with mininal effort.

The porosity of an HA film is fundamental to the success of the coating in vivo, and it should be on the few hundred nm to micron scale [54,55]. SEM imaging was performed on the samples of the HA film deposited with solutions 1–7 after 2, 4 and 7 process runs (Appendix A). A subset of these SEM images is included in Figure 3 to assess their mineral coverage and porosity. From Figure 3a,b, it can be concluded that solution 1 generated little or no coverage of mineral, with its topography being unchanged compared to the blank substrate after 7 process runs; see Appendix A.

Solution 4 showed the beginning of a porous network after 2 process runs, with mineral clusters forming a porous layer amidst needle-like interconnectivity; see Figure 3c. After 7 process runs, solution 4 shows thorough coverage and porosity on the sub-micron scale. Solution 7 shows a large build-up of mineral after 2 process runs without any apparent needle-like interconnectivity; see Figure 3e. After 7 process runs, solution 7 has good film coverage, and although pores can be seen, morphological features are larger, flatter and show evidence of cracking typical of poor adhesion. When compared with lower concentration solutions, it can be inferred that this solution results in a coating of high coverage but low porosity/high density. We postulate that this is related to the larger particle size of solution 7; particles should be on the nanometres scale to nucleate, and they should form a film with sub-micron features.

EDX analysis enhanced SEM observations. Lower titanium signals indicated higher HA coverage, and it can be seen in Figure 4a that the more concentrated solutions 5–7 show <5 at% titanium after just 2 process runs, but at lower concentrations, the HA layers are thinner. Solution 7 had no titanium detection in all post 7 process runs. Calcium and phosphorus signals are minimal for solutions 1 and 2. Solutions 3 and 4 show clear increases in these elements with process runs, while solutions 5–7 show greater than 12 at% Ca and 10 at% P following 2 process runs; see Figure 4b,c. Oxygen is 50–60 at% for all the samples, which is consistent with the presence of oxide-containing phases; see Figure 4d. Surface-bound carbon is universally observed across all samples, which is a typical occurrence with hydroxyapatite (HA) due to its well-known adsorption properties [56]. Notably, the carbon content exhibits an upward trend with increasing solution concentration, which is attributed to the greater availability of apatitic minerals for adsorption, as illustrated in Figure 4e. Figure 4f shows the NaCl content of the films. The NaCl concentration is small for all samples, regardless of the ionic strength (Na at% < 2.5% and Cl at% < 2%), which is consistent with the expectations because of the high solubility of NaCl. It is clear from these data that NaCl only acts to increase the ionic concentration and improve the availability of other ions for the precipitation process.

It was possible to quantify the amount of film deposited from the weight of the samples after each process runs; see Figure 5a. Mass uptake for solutions 1 and 2 is minimal. There is a linear trend of mass uptake per run with increasing concentration. Solutions 4–7 have an increasing intake of 1 mg per run, with solution 7 having 4 mg of film attachment per run.

The calcium to phosphate at% ratio (Ca/P) and the oxygen at% are widely used to identify hydroxyapatite phases; see Table 2 with data listed for relevant phases [57,58]. From Figure 5b, it is seen that there is generally a decrease in the Ca/P ratio as the number of process runs increases (where deposition is significant: solutions 3–7). Solutions 1 or 2 show the highest Ca/P ratios, and the data suggest that calcium attachment is preferential in the early stages of film development (nucleation); this is consistent with the substrate having negatively charged hydroxyl groups that attract Ca^2+^ from the solution. For solutions 1 and 2, there are not enough available ions to form TCP or OCP mineral as per Equations (1) and (2). After 2 process runs, both solutions 3 and 4 show a higher Ca/P, aligning with Equation (2), since a TCP content would increase the Ca/P. However, as deposition progresses (4, 7 runs), the OCP phase dominates; see Figure 5b. Figure 5b shows that solutions 3–7 are very closely aligned near the level of OCP in terms of Ca/P after 7 runs. Solutions 3–7, however, have an oxygen at% > 40%, implying that ACP is also present since ACP can have a flexible oxygen at%.

Together, these data sets reveal that solutions 1 and 2 had minimal apatite film attachment across all process runs, implying that their concentration and ionic strength are too low. Solutions 3 and 4 showed slower apatite film growth kinetics, but they showed the most adhered layer with a suitable mg of material added per run; also, they had the best morphological development after 7 process runs, mirrored through EDX data of the relevant elements at 2, 4 and 7 process runs. Compared to other studies, this morphology and phase composition are advantageous since the presence of TCP can accelerate coating degradation, whereas the mix of ACP and OCP allows for ease of integration and the structure encourages bone growth [60,61]. Solutions 5, 6 and 7 had large-scale apatite film attachment after only 2 process runs, but morphologically, emergence of some cracks were observed after 7 process runs with materials having fewer interconnecting needles and pores. Solutions 6 and 7 also exhibit high carbonate contents, corresponding to HA-based materials. The chemical composition of HA film from solutions 3–7 was matched at 7 process runs, indicating an OCP crystalline phase with ACP material. It is clear from this data that by altering the concentration and ionic strength of solutions, it is possible to expediate or remediate HA growth. However, growth follows the same nucleation and growth mechanism. Further detailed investigation of the crystallinity of HA phases is carried out in Part 2 of this work, including XRD analysis after various process runs to identify the growth mechanism of OCP, HA and ACP as they emerge from this solution process.

## 4. Conclusions

This study has shown that HA film can be deposited and controlled using a colloidal solution process. The repeatability of the process at 47 °C is confirmed through consistent dropping of pH in 30 min runs. Bonding of calcium ions to the substrate leads to a drop in pH, as explained in the proposed chemical pathway. DLS data show various particle sizes in solutions, with ~100 nm particles being preferred for porous and coherent films (because of the denser packing). SEM topography showed that various solutions used gave rise to a morphologically porous and chemically consistent HA coating. The concentration levels of solutions influence film thickness, coverage, and coating rate; NaCl was shown to have no role in HA bonding. Lower concentration solutions formed very thin films, with calcium and phosphate contents being <5 at% after 7 deposition runs. While some films contained carbonates at higher concentrations, all films had similar oxygen content. Formation of HA films at >1 mg per run is a strong indication that the phase of HA formation is consistent in all solutions. Ca/P ratio data strongly indicated an OCP phase; additionally, the oxygen atomic percentage indicated the presence of water and ACP. This HA phase aligns with the proposed nucleation and growth pathway from Equations (2)–(5).

Based on our results, it can be inferred that the optimal process solution is between solutions 3 and 4, as they clearly followed the proposed chemical nucleation and growth process. The advantage of this slower crystal growth is that it allows for excellent porosity and morphology, which can be seen via SEM, and the adhesion of the layer can be also be conveniently ratified with the photographs.

Overall, our study has shown that colloidal solutions can be used as a novel HA coating process, wherein crystallisation occurs in a timely manner at low temperatures.

## Figures and Tables

**Figure 1 nanomaterials-13-02577-f001:**
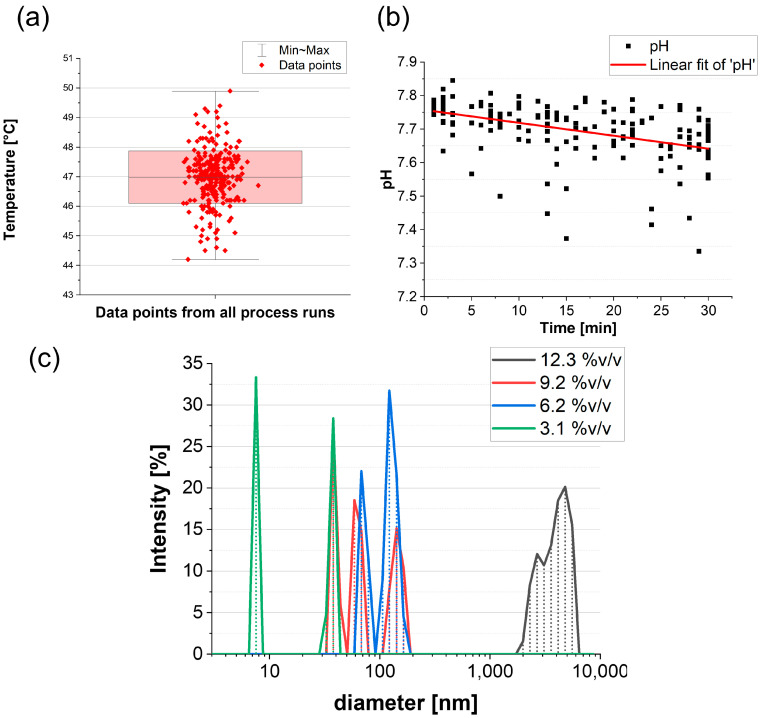
(**a**) Collated temperature data from all process runs, (**b**) pH data from all runs graphed against process time fitted with linear line of best fit and (**c**) dynamic light scattering (DLS) data of different solutions, intensity versus particle size.

**Figure 2 nanomaterials-13-02577-f002:**
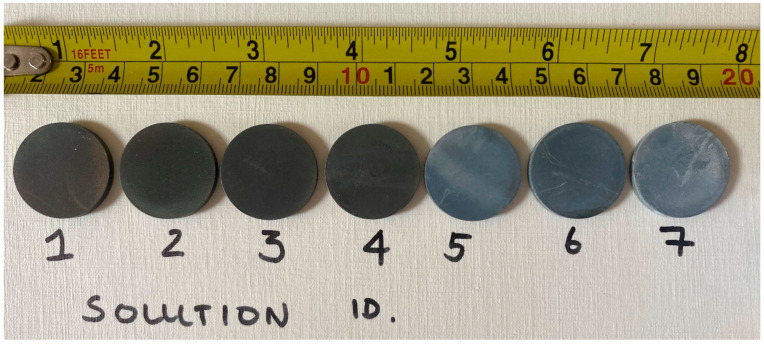
Photograph of HA samples on Ti coupons after being deposited with solutions 1–7.

**Figure 3 nanomaterials-13-02577-f003:**
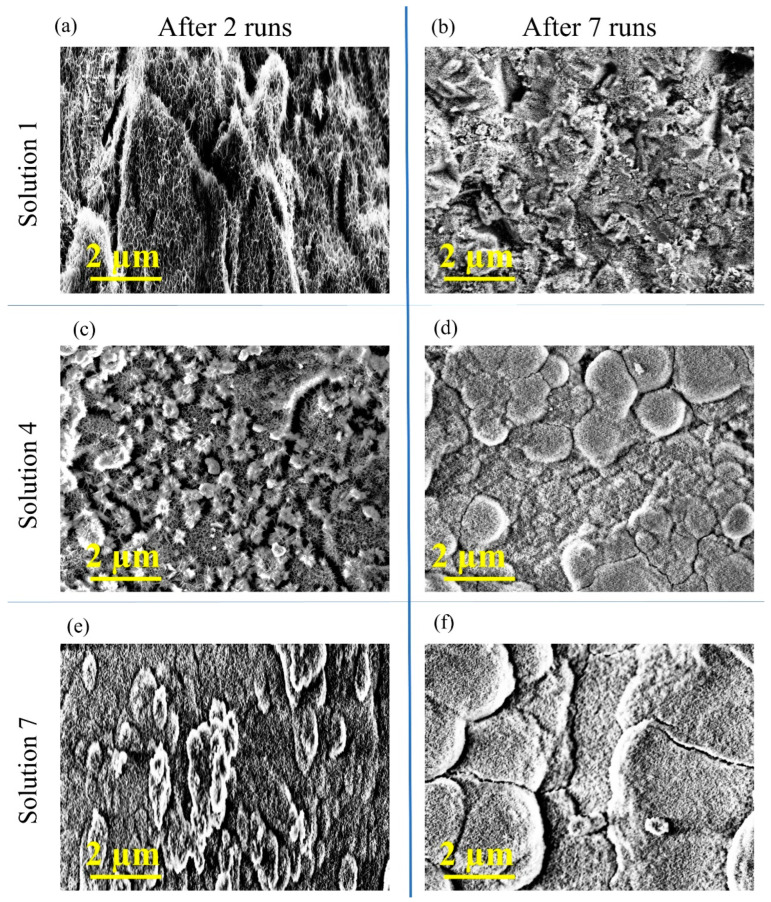
Scanning electron micrograph images of hydroxyapatite mineral formed on Ti-6AL-4V alloy. (**a**) image of the mineral deposited using solution 1 after 2 process runs, (**b**) image of the mineral deposited using solution 1 after 7 process runs, (**c**) image of the mineral deposited using solution 4 after 2 process runs, (**d**) image of the mineral deposited using solution 4 after 7 process runs, (**e**) image of the mineral deposited using solution 7 after 2 process runs and (**f**) image of the mineral deposited using solution 7 after 7 process runs.

**Figure 4 nanomaterials-13-02577-f004:**
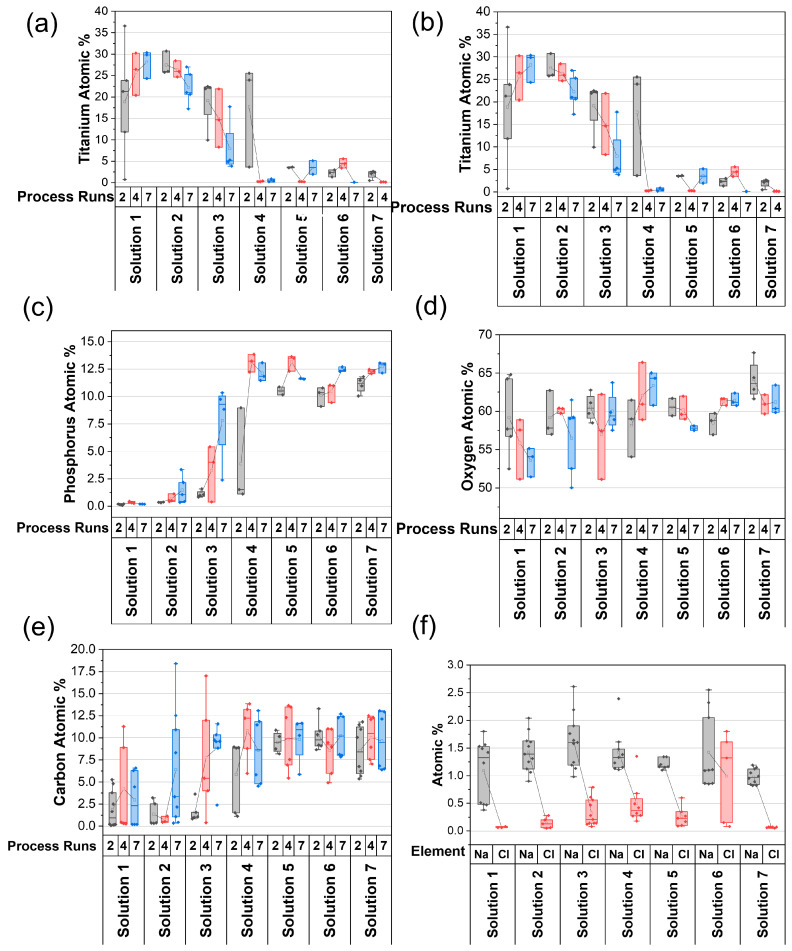
Energy-dispersive X-ray spectroscopy (EDX) spectra of deposited films recording various elements pertinent to hydroxyapatite deposition. All data points are the mean of 3 areas within a sample, grouped on the *x*-axis by solution ID, 1–7, and number of processes runs complete for each film (2, 4 or 7). ((**a**): titanium atomic %, (**b**): calcium atomic %, (**c**): phosphorus atomic %, (**d**): oxygen atomic %, (**e**): carbon atomic %, (**f**): sodium and chlorine atomic %).

**Figure 5 nanomaterials-13-02577-f005:**
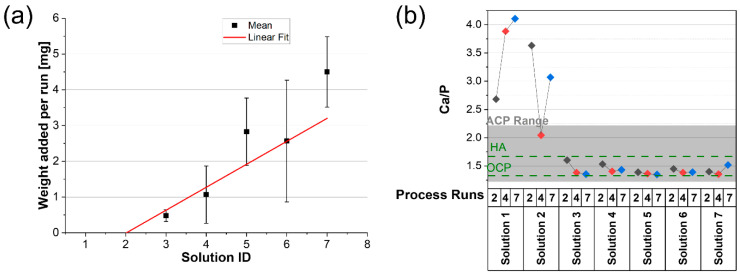
(**a**) Graph of weight added per run in mg versus solution ID and (**b**) the calcium to phosphate ratio (Ca/P) of films calculated for each solution after 2, 4, and 7 process runs.

**Table 1 nanomaterials-13-02577-t001:** Table containing list of process solutions used (1–7) in this work, with their corresponding percent solution to concentrate (%*v*/*v*) and ionic strength values (mmol·L^−1^).

ID	Concentrate to Solution Dilution [%*v*/*v*]	Ionic Strength [mmol·L^−1^]
Solution 1	3.1	163.08
Solution 2	4.6	244.71
Solution 3	6.2	326.81
Solution 4	7.7	407.71
Solution 5	9.2	489.74
Solution 6	10.8	570.76
Solution 7	12.3	652.36

**Table 2 nanomaterials-13-02577-t002:** Tabular data of hydroxyapatite phases: their acronym, mineral name, stoichiometric formula, calcium to phosphate ratio (Ca/P) and stoichiometric atomic percentage of oxygen [59].

**Acronym**	**Mineral**	**Formula**	**Ca/P Ratio**	**Atomic % Oxygen**
HA	Hydroxyapatite	Ca_10_(PO_4_)_6_(OH)_2_	1.67	34.9
ACP	Amorphous calcium phosphate	Ca_x_H_y_(PO_4_)_z_∙nH_2_O	1.2–2.2	Flexible
OCP	Octacalcium phosphate	Ca_8_H_2_(PO_4_)_6_·5H_2_O	1.33	39.7
TCP	Tricalcium phosphate	Ca_3_(PO_4_)_2_	1.5	41.3
CDHA	Calcium deficient hydroxyapatite	Ca_9_(HPO_4_)(PO_4_)_5_OH	1.5	43.13

## Data Availability

Data is contained within the article or Appendix A.

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
