# Peer review of "Development of Hydroxyapatite Coatings for Orthopaedic Implants from Colloidal Solutions: Part 1—Effect of Solution Concentration and Deposition Kinetics"

_nanomaterials, 2023, doi:10.3390/nano13182577_

Round 1

Reviewer 1 Report

The authors prepared hydroxyapatite coatings for orthopaedic implants with the solutions of calcium and phosphate ions. The effect of solution concentration and deposition kinetics was investigated. Comments:

1. The background and significance of coating should be carefully discussed in Introduction.

2. Why 47 °C was used for the stability investigation?

3. The conclusion should be shortened and refined.

4. The reference format should be revised.

Moderate editing of English language required.

Reviewer 2 Report

The authors report the effect of solution's stoichiometry and deposition cycles on the fabrication of hydroxyapatite coatings. I find the manuscript interesting and well-written, so publication can be recommended after addressing the following comments: 

1) "Chemical precipitation is advantageous at forming HA nanoparticles and nano-powders thanks to an ability to control the phases of HA but has not yet been extensively explored as coating technique.[15,16]" ; "Overall, this body of work had outlined that colloidal solutions can be used as a novel HA coating process whereby crystallising happens in a timely manner at low temperatures." - well these two sentences are not true, because the chemical precipitation is a well-developed technique for the deposition of HA. Please see: https://www.sciencedirect.com/science/article/pii/S1005030213002387?via%3Dihub ; https://link.springer.com/article/10.1007/s10973-019-08255-z (more literature can be found in the scientific databases). From that point-of-view, the novelty of your work is doubtful. More scientific input is needed to describe the innovation in your study. 

2) Nearly 90% of cited references are outdated (published before 2019). Moreover, there are many references from early 00s and even 1980s, 1960s, which could be accepted as a solid evidence for lack of novelty and current scientific interest in this particular field. It is strongly recommended to replace almost all of the cited literature with sources published within 2019-2023. Without such a correction, the manuscript will not meet the standards of Nanomaterials journal. 

3) Materials and methods - what is the water solubility of used chemicals and how much of each is added in the de-ionized water. What is the volume of the latter? 

4) The findings of your research are revealed without any interpretation of the physical and chemical processes happening in the working solutions and then at the solid interface. Also, there is no comparison of the morphochemical peculiarities of your coatings with other existing counterparts. These mistakes must be fixed prior to publication. 

5) Hydroxyapatite has antibacterial properties, so it will be nice to elaborate upon the possible practical applicability of this material. Recommended research: https://www.nature.com/articles/s41598-019-40488-8 ; https://www.sciencedirect.com/science/article/pii/S0264127520303245 

6) Supporting information file is provided, but its content is not mentioned in the main text.

The English language and style are fine. 

Reviewer 3 Report

The manuscript is devoted to the preparation of hydroxyapatite (HA) coatings by deposition from colloidal solutions. This is a rare method that is not commonly used, apparently, due to limited success in the preparation of coatings with desired properties. Although, the authors do not seem to be quite successful either, this study is interesting and can be useful for further improvement of this method. So, I believe that the manuscript could be accepted for publication in Nanomaterials after a major revision addressing the following issues.

  1. Equation 1. The formula for calculation of the ionic strength presented here is incorrect. Ionic strength is a half-sum of the ion concentrations multiplied by the square of their charges. It is not clear what the letter S here stands for and the concentrations should not be taken as squares. I hope that the authors did use a correct formula in their calculations of the ionic strength. However, please, check if my assumption is correct. If it is not, please correct the results of the calculations.
  2. Equation 5 does not seem to be balanced. There are 6 P atoms at the left and only 4 at the right. Oxygen and hydrogen atoms are not balanced either. Something is definitely missing here.
  3. Line 169. Please explain in more detail why there is no precipitation below 44 °C?
  4. Figure 1a. It is not clear what is at the X axis of this graph.
  5. Figure 1b. It is written in the text that there was not change in pH after 30 minutes. Please, show this fact at the graph.
  6. Line 199. It is not clear what the phrase: “…this mineral is deposited rather than highly adhered crystal” means. The following phrase “growth like from solutions 3 and 4 which are darker” is also not clear. Please rephrase and/or clarify.
  7. Line 231. The carbon concentration in the samples seems to be quite high, exceeding 10% after the HA deposition. Could you explain its state and origin rather than just calling it “adventitious”? Is it some sort of carbonate/hydroxocarbonate with CO2 coming from the air? In fact, the presence of carbonates is stated in Conclusions, but not in the main text.
  8. Figure 5a. Could you estimate the average thickness of the film corresponding to added weight?
  9. Line 259. I failed to find information how the samples were dried after the HA deposition. High oxygen concentration may imply the excess of adsorbed water that was not properly removed from the surface by drying.
  10. Paragraph starting at Line 269. Does HA in the meaning expressed in this paragraph also include OCP and ACP? If it does not as they are different chemical compounds, the authors should find some way to quantify the amount of deposited HA. Otherwise, the conclusions made in this paragraph do not seem to be properly supported by the experimental data. If HA is assumed here to include OCP and ACP, this fact should be clarified in Section 3.1. Currently, after reading this Section I understood that these were different species.
  11. XRD data that are important for identifying the obtained phases are notably missing from the manuscript. As this is Part 1 of this title, these data might be included in Part 2. If so, some text clarifying this fact and addressing the reader to Part 2 should be included. Otherwise, there is no direct confirmation that HA was actually synthesized.

Round 2

Reviewer 1 Report

Accept in present form

Author Response

We thank the reviewer for their lovely response

Reviewer 2 Report

By comparing the text in the Introduction section, one may see that there is very slight difference between the original and revised version, although the highlighted text (which is supposed to reflect the CHANGES) is a large part of the Introduction. An example:    Original text: " Orthopaedic implant fixation is hindered by fibrosis encapsulation which is the body’s natural response to a foreign object.[1–5] Coating metallic implants with a bio-compatible layer can mitigate fibrous build-up and promote implant fixation.[6] Being the main ceramic component of bones and teeth in the human body and possessing rigidity, compressive strength and a brittle nature, hydroxyapatite (HA) coatings are used to improve the osteointegration of orthopaedic implants."   Highlighted text: " Orthopaedic implant fixation is hindered by fibrosis encapsulation which is the body’s natural response to a foreign object.[1–4] Coating metallic implants with a bio-compatible layer can mitigate fibrous build-up and promote implant fixation.[5,6] Being the main ceramic component of bones and teeth in the human body and possessing rigidity, compressive strength and a brittle nature, hydroxyapatite (HA) coatings are used to improve the osteointegration of orthopaedic implants."   In other words, the authors have highlighted large fractions in the text (likely with the aim to convince the reviewers that they have performed serious revision), but the real modifications are minor!   This is just an example of the lightly performed revision, which leaves me no other choice unless recommending rejection.

English language and style are fine.

Reviewer 3 Report

All my comments from the original review have been addressed. So, the manuscript can be accepted for publication.

Author Response

(The authors gave the same response as above.)

Round 3

Reviewer 2 Report

Ok.

The English language and style are fine.